# Biogeochemical Factors of Cs, Sr, U, Pu Immobilization in Bottom Sediments of the Upa River, Located in the Zone of Chernobyl Accident

**DOI:** 10.3390/biology12010010

**Published:** 2022-12-21

**Authors:** Darya Zelenina, Natalia Kuzmenkova, Denis Sobolev, Kirill Boldyrev, Zorigto Namsaraev, Grigoriy Artemiev, Olga Samylina, Nadezhda Popova, Alexey Safonov

**Affiliations:** 1A.N. Frumkin Institute of Physical Chemistry and Electrochemistry, RAS, Obrucheva Str. 40, Moscow 117342, Russia; 2Radiochemistry Division, Faculty of Chemistry, Lomonosov Moscow State University, Leninskie Gory, Moscow 119991, Russia; 3V. Vernadsky Institute of Geochemistry and Analytical Chemistry, RAS, Kosygina Str. 19, Moscow 119991, Russia; 4Nuclear Safety Institute, RAS, Bolshaya Tulskaya St. 52, Moscow 115191, Russia; 5Kurchatov Centre for Genome Research, NRC Kurchatov Institute, Akad. Kurchatov Sq., 2, Moscow 123098, Russia; 6Winogradsky Institute of Microbiology, Research Centre for Biotechnology, Russian Academy of Sciences, Prospect 60-Letiya Oktyabrya 7/2, Moscow 117312, Russia

**Keywords:** freshwater bodies, bioremediation, phytoplankton, Cs, Sr, U, Pu, sediments, biomineralization

## Abstract

**Simple Summary:**

Strong fixation of Sr, U, and Pu in river bottom sediments can occur under conditions of sufficiently high N, P, S concentrations. In consequence, this environment facilitates active phytoplankton growth and the formation of biogenic minerals. These conditions can be achieved by adding the necessary nutritive components to the water and bottom sediments of polluted water bodies.

**Abstract:**

Laboratory modeling of Cs, Sr, U, Pu immobilization by phytoplankton of the river Upa, affected after the Chernobyl accident, has been carried out. Certain conditions are selected for strong fixation of radionuclides in bottom sediments due to biogeochemical processes. The process of radionuclide removal from the water phase via precipitation was based on their accumulation by phytoplankton, stimulated by nitrogen and phosphorus sources. After eight days of stimulation, planktonic phototrophic biomass, dominated by cyanobacteria of the genus *Planktothrix*, appears in the water sample. The effectiveness of U, Pu and Sr purification via their transfer to bottom sediment was observed within one month. The addition of ammonium sulfate and phosphate (Ammophos) led to the activation of sulfate- and iron-reducing bacteria of the genera *Desulfobacterota*, *Desulfotomaculum*, *Desulfosporomusa*, *Desulfosporosinus*, *Thermodesulfobium*, *Thiomonas*, *Thiobacillus*, *Sulfuritallea*, *Pseudomonas*, which form sulphide ferrous precipitates such as pyrite, wurtzite, hydrotroillite, etc., in anaerobic bottom sediments. The biogenic mineral composition of the sediments obtained under laboratory conditions was verified via thermodynamic modeling.

## 1. Introduction

Accidents occurring during nuclear fuel production and processing or active nuclear power plant operation, threaten subsequent radionuclide contamination of surrounding territories. Historically, the most striking accidents of this nature include the Three Mile Island accident (1979) (Voelkle, 2015) [1], Fukushima-Daiichi (2011) [2], and Chernobyl (1986). The Chernobyl accident contaminated an area measuring approximately 60,000 km^2^, including the Bryansk, Kaluga, Orel, and Tula regions of Russia, and significant areas of the Republic of Belarus (about 46,000 km^2^) and Ukraine (about 38,000 km^2^) [3,4]. Other accidents have occurred at nuclear fuel fabrication and reprocessing facilities such as the Mayak Plant, Russia (1948–1949) [5]; the Choke River Laboratory, Canada (1952) [6]; the JCO radiochemical plant in Tokaimura, Japan (1999) [7]; etc. The greatest hazard resulting from these accidents was the release of long-lived actinides (U, Pu, Np, Am), as well as intermediate-lived ones (^90^Sr and ^137^Cs) [8]. Soil contamination may be effectively cleaned by soil upper-layer removal, phytoremediation, or by reagent treatment. However, these methods are not applicable for the treatment of polluted water bodies. At the same time, in contaminated water bodies, there are more significant risks of radionuclides spreading and entering ecological food chains. There is a high risk of radionuclide spread when surface natural or artificial reservoirs, such as the Techa River and Lake Karachay at the Mayak Industrial Site, are used as radioactive waste repositories. For this reason, such water bodies are subject to remediation and conservation in accordance with modern safety requirements [9].

The behavior of synthetic radionuclides in water bodies is influenced by their physicochemical characteristics, such as oxidation state, chemical forms in solution (true solubility, solubility in the form of organic or mineral complexes, or in the form of colloidal or pseudocolloidal particles) [10,11]. One should also take into account the factor of radionuclide migration in water bodies due to hydrological [12] and wind-wave regime [13], bottom topography [14], depth [15], and physicochemical characteristics such as temperature–oxygen regime [16], pH value, and reduction potential (*Eh*) [17,18], salt content [15], etc. Radionuclides may bind with suspended particles to form precipitates that can subsequently become fixed in bottom sediments. Thus, under favorable conditions, the reservoir water exhibits a self-purifying mechanism [19]. Under unfavorable conditions, radionuclides remain in the water phase or undergo constant migration between the water phase and bottom sediments.

An important process during the self-cleaning of the water body occurs in the summer during periods of intensive development of phytoplankton, the so-called “water body bloom” [20]. Other biological factors also have a strong influence, e.g., the taxonomic and functional diversity of microflora, phyto- and zoocenosis [21]. In the water column, a large amount of biomass comprising photosynthetic macro- and microalgae and cyanobacteria acts as an analogue of the biosorbent with a large surface area [19,22,23,24,25]. Further, following the death and subsequent sinking of the cells, radionuclides in the bottom sediments begin to accumulate and increase in concentration [26]. The efficiency of biogeochemical self-purification processes is determined by the mechanism of radionuclide immobilization in bottom sediments (biosorption, bioaccumulation in cells, or biomineralization in new mineral phases) [27]. Biosorption of radionuclides occurs on the surface of organisms [28] or via accumulation inside the cell. After cell lysis, if immobilization in biogenic mineral phases has not occurred, there exists a high probability of metal remobilization [29]. It is well known that during the development of anaerobic sulfate-reducing or iron-reducing microorganisms, insoluble precipitates of pyrrhotite (Fe_n_S_n+1_) [30], troilite (FeS) [31], and hydrotroilite (mFeS×nFe (OH)_2_) [32] are formed. The growth of the aerobic bacteria may lead to the formation of ferrihydrite (mFe_2_O_3_ × nH_2_O) [33], goethite (FeOOH) [34], and other precipitates [35,36], which promote actinide mineralization [37]. Thus, to predict the self-purification of a water body, a different set of parameters is required for the biogeochemical modeling of these processes.

Bioremediation is widely used for the highly effective immobilization of radionuclides in soils and rocks of aquifers [38,39,40,41,42], and this approach can be applied to create a biogeochemical barrier for radionuclides in bottom sediments. Understanding the mineral formation processes in bottom sediments and their role in the immobilization of radionuclides, as well as developing approaches for their management, could be an important stage in the development of new, effective approaches for clearing water bodies of radionuclide contamination.

The aim of this work is to experimentally verify and simulate radionuclide deposition during phytoplankton stimulation in a microcosm obtained from the River Upa (Tula Oblast) affected by the Chernobyl accident.

## 2. Materials and Methods

### 2.1. Description of the Site

Water (5 L) and bottom sediment samples were taken in November 2019 from the Upa River backwaters (Figure 1) in the Tula region between the village of Krapivna and the village of Orlova (53.954297° S, 37.158518° E). The Úpa River basin (9500 km^2^), is located in the northern part of the Central Russian Upland. The river itself drains the central and southern parts of the Tula region. The content of radionuclides, accumulated in floodplain sediments after the Chernobyl fallout, reflects changes in the concentration of ^137^Cs in the suspended sediments carried by the river, both over time and along the Upa valley. It should be noted that the sediment load, formed in the upper, rather heavily contaminated part of the Upa River basin, is almost completely intercepted by the Shchekinskoye storage reservoir. The main volume of radionuclides entering the Upa downstream of the Shchekinskoye storage reservoir is linked to the sediment load of its southern left-bank tributaries, the largest of which is the Plava [43,44,45].

The river drainage basin is located in a cold temperate climate at an elevation of 159 m above sea level. According to the Koppen–Geiger climatic classification, the studied section of the Upa River basin is located in the humid continental (Dfb) climatic zone characterized by dry seasons and hot summers [46]. The annual precipitation is around 703 mm. The difference in precipitation between the driest and the wettest month is 45 mm. The average annual temperature is 5.9 °C. The temperature varies by 27.8 °C throughout the year [47].

### 2.2. Experiment

#### 2.2.1. Cultivation of a Phototrophic Enrichment Culture

The cultivation was carried out at room temperature for 30–45 days under constant illumination with a halogen lamp at a light intensity of 232 lx on BG-11 medium, (Hughes et al. (1958) [47], further modified by Allen (1968) [48], Rippka et al. (1979) [49]. Composition of BG-11 medium (g/L): NaNO_3_ (1.5); K_2_HPO_4_(×3H_2_O) (0.04); MgSO_4_ × 7H_2_O (0.075); CaCl_2_ × 2H_2_O (0.05); Na_2_CO_3_ (0.02); citric acid (0.006), EDTA-Na_2_ (0.001); Fe^3+^/NH_4_^+^-citrate (brown, 16–19% Fe) (0.006); micronutrient solution (1 mL).

The biomass of phytoplankton enrichment culture for the experiments was obtained by stimulating growth in 2-L flasks and then separating it from the medium by centrifuging the cells at 6000 rpm for 5 min.

#### 2.2.2. Determining the Effect of Mineral Additives on Accelerating Biomass Growth

In total, six 20-mL water samples were used, each containing 200 µL of concentrated mineral additive solution at the concentrations listed below. The mineral additives for phytoplankton stimulation were as follows (mg/10 mL): urea (43) + sodium sulfate (20); ammonium nitrate (53) + sodium sulfate (20), potassium nitrate (72) + sodium sulfate (20), ammophos (Ap) (100) + sodium sulfate (20), potassium hydroorthophosphate (45) + sodium sulfate (20), urea (43) + potassium hydroorthophosphate (45) + sodium sulfate (20). All experiments were performed in triplicate.

The duration of the experiment was 67 days (until complete settling of the cells in the bottom sediments) under constant light (232 lx) and at room temperature ranging between 21–23 °C.

#### 2.2.3. Assessment of Radionuclide Immobilization Efficiency in Phytoplankton Development

The experiment was carried out in polypropylene vials with 50 mL of Upa river water, where the radionuclide was added. Solutions containing 1 mL of nitric acid and the following radionuclides were used for addition: specific activity in the sample (Bq/L): ^233^U—8310, ^239^Pu—8334, ^137^Cs—8360, ^90^Sr—8290.

Aliquots of liquid phase were taken at intervals (1 and 24 h, 14, 30, 60, 90, 120 and 150 days) for radionuclide analysis. The radionuclide sorption effectiveness (S, %), and the corresponding distribution coefficient (Kd) were calculated by the formula:$$K_{d}=\frac{C_{sph}}{C_{lph}}\cdot \frac{V}{m}$$
where *C_sph_* (Bq/L) is the radionuclide activity in the solid phase; *C_lph_* (Bq/g) is the radionuclide activity in the liquid phase; *V* is the volume of the liquid phase, cm^3^; *m* is the mass of the solid phase, g.

To assess the binding strength of sorbed radionuclides, a one-stage desorption using the same lake water was carried out after the sludge sample had dried. Therefore, 100–200 mg of each sample was placed in vials and stirred for 2 h.

#### 2.2.4. Modeling Radionuclide Behavior in Sludge

To determine the behavior of radionuclides in sludge, 5 samples of 15 mL of model water were taken, and sludge from the model object and an anaerobic bacteria stimulation additive were added to each sample. The experiment was carried out in hermetically sealed 25 mL penicillin vials. As an additive for enhanced biomass growth of chemotrophic bacteria, 0.5 mL of Ap at a concentration of 60 mg per 10 mL was used. Ap was combined with a sulfur source, Na_2_SO_4_ (0.05 mL). The effect of Ap and sulfur sources separately was also considered. At certain time intervals, sample pH and Eh values were measured.

### 2.3. Analytical Methods

>Macronutrient concentrations were determined using a Capel-205 capillary electrophoresis system of the latest generation. Identification and quantification of analyzed cations and anions were carried out via the indirect method of measuring UV-absorption at 254 nm. Electrophoresis was carried out in untreated fused silica capillaries, 60 cm long (effective length—50 cm), and 75 µm inner diameter. The capillary was incubated at 20 °C with an applied voltage of +13 kV for cations or −17 kV for anions.

The content of trace elements was determined using an iCAP Qc inductively coupled plasma mass spectrometer (Thermo Scientific, USA) and an Optima-4300 DV inductively coupled plasma atomic emission spectrometer (Perkin-Elmer, MA, USA).

Radionuclide content was determined via liquid scintillation using the Tri-Carb-3180 TR/SL radiometer (PerkinElmer, Waltham, MA, USA).

The pH value was determined using a pH-meter-ionometer pH-150MI (Izmeritelnaya technica Moscow, Russia), equipped with a combined electrode. The Eh value was determined using a multifunctional device “Expert-001” (Econix-expert, Moscow, Russia) equipped with a combined electrode.

Electron microscopy was performed using a TESCAN MIRA3 FEG-SEM scanning electron microscope from the Joint Use Center, Vernadsky Institute of Geochemistry and Analytical Chemistry, Russian Academy of Sciences. For SEM analysis, samples were placed on an aluminum holder with an electrically conductive tape, and vacuum carbon deposition (Q150T E Plus) (vacuum 4^−3^, current 50 A) was performed. The samples were captured in two modes, SE and BSE, at a voltage of 20 kV.

Elemental analysis of materials was conducted using an Axiosm AX Advanced wavelength-dispersive X-ray spectrometer (manufactured by PANalytical, Netherlands). Milled samples <0.1 mm in size were air-dried at a temperature of 383 K. The determination of loss of ignition (LOI) was conducted by gravimetric method at a temperature of 1273 K. Glass disks for determination of composition were prepared by induction melting of annealed sample powders mixed with Li_2_B_4_O_7_ at 1473 K.

The dry cell mass determination was analyzed by separating the biomass from the medium on Millipore filters with a pore diameter of 0.22 μm. The filters were brought to a constant weight in an oven at 80 °C, weighed, and then 10 mL of culture suspension with a determined optical density was filtered using a vacuum pump. The filters were again brought to a constant weight, and the difference was used to determine the mass of dry cells in mg/mL.

The centrifugation of biomass for separation from solution was carried out on an OPN-8m centrifuge at a rotor speed of 6000 rpm.

Visual analysis of the algal diversity was performed using a JENAVAL Carl Zeiss microscope equipped with a Zeiss Bundle Canon PS G9 digital camera (Zeiss, Jena, Germany) at magnifications of ×400 and ×1000.

DNA was isolated using the ZymoBIOMICS™ DNA Miniprep Kit (Zymo Research, Irvine, CA, USA) according to the manufacturer’s instructions. Variable regions of the 16S rRNA gene in the V3–V4 region were selected for amplification in the preparation of libraries.

DNA amplification in all cases was carried out by real-time PCR on CFX96 Touch (Bio-Rad, Hercules, CA, United States) with the qPCR mix-HS SYBR reaction mixture (Evrogen, Moscow, Russia). In preparing libraries for amplification, variable parts of the V3–V4 region of the 16S rRNA genes were selected. For amplification of the V3–V4 region, the degenerate primers For341 (5′-CCTACGGGNBGCASCAG-3′) and Rev806 (5′-GGACTACHVGGGTWTCTAAT-3′) were used. For amplification of the V4 region, the degenerate primers For515 (5′-GTGBCAGCMGCCGCGGTAA-3′) and Rev806 (5′-GGACTACHVGGGTWTCTAAT-3′) were used. Amplification was performed by real-time PCR on a CFX96 Touch (Bio-Rad, Berkeley, CA, USA) using the qPCR mixHS SYBR (Eurogen, Moscow, Russia). Denaturation, primer annealing and chain elongation for regions V3-V4 were performed at 96, 54 and 72 °C, respectively. In addition, the steps for region V4 were performed at 96, 58 and 72 °C, respectively. Purification of the desired product from each batch was carried out using Agencourt AMPure XP magnetic particles (Beckman Coulter, Brea, CA, USA). In addition, high-throughput sequencing was performed using a MiSeq system (Illumina, San Diego, CA, USA) and a reagent kit (MiSeq Kit v2, 500 cycles, Illumina, CA, USA). BioSample accessions numbers: SAMN31891071, SAMN31891072

### 2.4. Geochemical Modeling

The PHREEQC 2.18 (pH-REdox-EQuilibrium) software [50] was used to determine the element speciation in solutions and saturation indices. The PHREEQC is based on the solution of systems of mass action law and material balance equations.

The saturation indices for the solid mineral phases of the studied radionuclides were assessed in modeling. The saturation index, SI, is the difference between decimal logarithms of current products of the ionic activity of the *i*-th phase and the solubility constant of the relevant mineral:$$SI_{i}=\mathrm{lg}IAP_{i}-\mathrm{lg}Ks_{i}$$

If the logarithm of SI exceeds 0.5, the formation of this phase may be predicted; if it is below—0.5, its dissolution is probable. In total, three bases of thermodynamic data (TDB) were used for geochemical modeling: llnl, PSINA, and NEA.

## 3. Results and Discussion

### 3.1. Sample Characteristics

#### 3.1.1. Chemical and Radiochemical Composition of Liquid Samples

The results detailing the major ions, trace elements and radionuclide contents detected in the samples are given in Table 1 and Table 2. The pH value of the water sample was 6.8, and the Eh value +95 mV.

The water sample shows low mineralization with a relatively high concentration of bicarbonate and chloride ions. Among the trace elements, high concentrations of zirconium, manganese, iodine, nickel and iron were observed. Notably, no technogenic radionuclides were found in the studied water samples. It is also important to note that the content of the main biophilic elements in the water sample from the Upa River was low. Thus, the content of iron did not exceed 3.2 mg/L, phosphorus 0.4 mg/L, potassium 11.8 mg/L, and the total nitrogen content did not exceed 2 mg/L.

#### 3.1.2. Chemical, Radiochemical and Mineral Composition of Sludge Samples

Gamma-spectrometric analysis of the bottom sediments obtained from the river Upa showed the presence of natural radionuclides ^40^K—3259 Bq/kg, ^232^Th—719 Bq/kg, ^226^Ra—197 Bq/kg, and regarding technogenic radionuclides, only ^137^Cs was found at a concentration of 109 Bq/kg. Other technogenic radionuclides were not detected.

Table 3 shows the elemental composition of the sludge sample. The organic matter content was about 70 wt %. The high organic-matter content caused problems during quantitative X-ray phase analysis of the sample. Based on the elemental composition, we can assume a presence of clay fraction, and oxide ferrous phases typical of freshwater-running water sludge in the bottom sediment samples.

### 3.2. Phytoplankton and Microbial Diversity

#### 3.2.1. Phyto- and Bacterioplankton Diversity in the Water Samples

The analysis of prokaryotic diversity in the Upa River water samples using high-throughput sequencing of 16S rRNA genes (Figure 2) showed the predominance of phyla *Cyanobacteriota* (41%), *Actinomycetota* (21%), *Pseudomonadota* (*Proteobacteria*) (16%) and *Bacteroidota* (7%). *Cyanobacteria* were dominated by representatives of the genus *Planktothrix* (order *Oscillatoriales*) which are filamentous cyanobacteria associated with algal blooms in aquatic ecosystems worldwide. Less than 1% of the 16S rRNA gene sequences belonging to genera *Cyanobium* and *Pseudanabaena* were found among cyanobacteria. Light microscopy performed on the water samples confirmed the molecular data (Appendix A), although it revealed a somewhat greater diversity of cyanobacteria: heterocystous nitrogen-fixing *Dolichospermum spiroides* and unicellular *Microcystis*, of which both were morphologically identified in the sample (Appendix A).

The rest of the bacterial diversity of water samples identified by high-throughput sequencing was represented by heterotrophic (organotrophic) microorganisms. Representatives of *Actinomycetota*, *Pseudomonadota*, and *Bacteroidota* utilize organic metabolites and polysaccharides produced by cyanobacteria, eukaryotic algae, and higher plants [51,52,53].

Light microscopy revealed representatives of the phyla *Chlorophyta* (green algae) and *Bacillariophita* (diatoms). Green algae were represented by the unicellular *Chlorella* and *Chloroidium* and the multicellular *Planctonema*, *Scenedesmus*, *Coelastrum*, and *Pediastrum*. Among *Bacillariophyta* representatives of centric diatoms belonging to genera *Skeletonema*, *Stephanodiscus*, and *Cyclotella*, as well as pennate diatoms of the genus *Synedra*, were most widely represented (Appendix A).

#### 3.2.2. Microbial Diversity of the Bottom Sediment

The microbial diversity analysis results of bottom sediment based on 16S rRNA gene sequencing are shown in Figure 3 and Appendix A. The sulfur cycle anaerobic microorganisms dominated the sample, including sulfate-reducing bacteria of the phylum *Desulfobacterota* (20%), representatives of the *Firmicutes* (including sulfate-reducing genera *Desulfotomaculum*, *Desulfosporomusa*, *Desulfosporosinus*, *Thermodesulfobium)* are known. A number of representatives of the classes *Alphaproteobacteria* (9%) and *Gammaproteobacteria* (19%) in the phylum *Pseudomonadota* (including *Thiomonas*, *Thiobacillus*, and *Sulfuritallea*) are capable of dissimilatory and assimilatory sulfate reduction, as well as oxidation of reduced forms of sulfur. It is important to note the presence of bacteria of the genus *Pseudomonas* in the bottom sediment sample. These bacteria are known for their ability to use a wide range of inorganic electron acceptors, including nitrate, arsenate, uranyl, and selenite [54,55,56]. Of the 16S rRNA sequences, 21% were attributed to mitochondria of eukaryotic aquatic organisms. No cyanobacterial genes were found, likely due to their destruction by the anaerobic microbial community upon ingestion into the bottom sediment.

### 3.3. Phytoplankton Activation by Additives

#### Biomass Accumulation and Number of Morphotypes

After adding mineral solutions containing nitrogen and phosphorus compounds to water samples, the development of planktonic phototrophic biomass was observed after an average period of 8 days. At the same time, only urea with potassium phosphate, Ap and potassium nitrate initiated the development of intensive green coloring of water samples over the 2-month period (the average vegetation period of intensive phytoplankton development for the Tula region). In the other cases, a minor effect was observed only at the beginning of the experiment, with no stable planktonic phototrophic community development.

The increase in phytoplankton biomass after the addition of the stimulants as well as the number of morphotypes are presented in Table 4.

In the water sample without additives, which the filamentous cyanobacterium *Planktothrix agardhii* was dominant. In the experiment with the additives, the nitrogen-fixing filamentous cyanobacteria of the genus *Dolichospermum* and the green microalgae of the genera *Chlorella* and *Scenedesmus* were dominant. The highest number of morphotypes was found in the variant with Pu addition (on average 5 morphotypes), a lower number was found in the variant with uranium (on average 4 morphotypes) and cesium (on average 3.5 morphotypes). The lowest number of morphotypes was detected in the variant with the addition of strontium (on average 2, morphotypes).

The addition of radionuclides led to a decrease in phytoplankton biomass accumulation (Table 4). In the control variant without the addition of radionuclides and nutrients, up to 20 mg/L of biomass was accumulated, while in the variants with radionuclides but without nutrients, only 10 mg/L was accumulated. The addition of Ap to the experiment system led to a 2–6 fold increase in biomass, and the addition of urea with phosphate resulted in a 5–12 fold increase in biomass compared to variants without access to nutrients. The greatest increase in biomass with nutrients was observed in the control variant without radionuclides. Phytoplankton growth was stimulated to a lesser extent in the variants with the addition of Pu, U, and Sr, and the lowest growth stimulation was observed in the presence of Cs.

### 3.4. Laboratory Modeling of Radionuclide Behavior in the Water-Bottom Sediment System

#### 3.4.1. Evaluation of Radionuclide Removal by Phytoplankton Activation with Stimulant Additives

Without the addition of growth promoters (Figure 4), the maximum removal rate was observed for plutonium (up to 90%), due to its ability to hydrolyze at pH 7 and to form suspensions [57]. Sr and U removal did not exceed 40%, and Cs removal was less than 10%.

During phytoplankton growth (Figure 4), the most significant changes were observed for uranium concentration in the liquid phase. When Ap was added, its immobilization efficiency exceeded 90%; in the system with urea and phosphate, more than 80% of the uranium was precipitated. When urea and phosphate were added, an almost two-fold increase in strontium immobilization efficiency was observed; the addition of Ap did not result in significant changes. In both experiments, a slight decrease in the removal efficiency of plutonium from the aqueous phase was observed (by no more than 15% for Ap stimulation and by no more than 10% for urea and phosphate addition). No significant cesium accumulation in the sludge was detected.

The results obtained can be explained both by the removal effect of the metals from the phytoplankton cells and by the effect of the phytoplankton-stimulating additives. It is known that uranium and strontium form insoluble phosphate precipitates [58,59]. Strontium can accumulate in cells in carbonate phases, similar to calcium [60].

The mechanisms of uranium extraction by phytoplankton are described in detail in [24]. The coefficient of uranium accumulation by different phytoplankton species ranges from 30–60 to 600–1600. Uranium is known to be sorbed onto the cells themselves, on their metabolic products, and to accumulate in the periplasm, and then reduced by bacterial cells [61,62]. Therefore, high accumulation coefficients in biomass are observed.

The lack of a positive effect of additives in removing plutonium can be explained by the formation of colloidal, soluble phosphate, or organic compounds. The removal of plutonium by cells generally follows the same mechanisms as for uranium [63].

The low efficiency of Cs removal is not contradicted by other researchers [19]. This is due to the movement of the cesium by passive and active transport (through the (Na^+^–K^+^) ion pump in the bacterial cell membrane) [64] As with other alkali metals, there are no biomineralization mechanisms for cesium in cells.

#### 3.4.2. Assessment of Binding Force of Radionuclides to Bottom Sludge

As a result of the previous experiment, sediment with radionuclides was obtained from the stimulated and no stimulated phytoplankton. Desorption of radionuclides (Figure 5) by river water sample from dried sediment showed that in the variants of the experiment with additives, plutonium, strontium and uranium changed to a less mobile form. In an experiment with unstimulated sediment, 75% of Cs, about 10% of Pu 23% of Sr and 37% of U were released into the aqueous phase during desorption.

The cesium yield from biomass was significant in all cases, reaching 80%, while the variant of the experiment with Ap showed an increase up to 100%, with the addition of urea and phosphate resulting in a decrease to 70%. The high activity of the microflora, which performs the biomass destruction, with accumulated cesium, can be used to explain the enhanced cesium yield in the experiment with Ap.

### 3.5. Thermodynamic Modeling of Radionuclide Behavior in Sludge

#### 3.5.1. Changing Physical and Chemical Conditions in Bottom Sediments

Thus, the strongest fixation of radionuclides in bottom sediments was observed under anaerobic conditions. We used 1 g/L sulfate ions in experiments to activate anaerobic bacteria from the sulfur and iron cycles, promoting the accumulation of secondary biogenic minerals. During the experiment, it was found that after the stimulation of phytoplankton in the sludge, there is an active microbial process of oxygen consumption, which leads to a decrease in the redox potential of the medium into the anaerobic zone (Figure 6). This process occurs most effectively when Ap, and Ap with sodium sulfate are added. It is worth mentioning that in the experiment without additives, anaerobic conditions also gradually formed when the vial was sealed.

#### 3.5.2. Thermodynamic Calculation of Radionuclide Forms in Bottom Sediments

The change in redox potential of the system under the influence of microbial processes has a significant influence on the distribution forms of the redox-sensitive radionuclides, in our case, which include uranium and plutonium. For detailed calculation of the main forms of radionuclides under geochemical modeling using thermodynamic constants from different databases are used.

Figure 7, Figure 8 and Figure 9 show the values of the saturation indices of the main phases, which were subjected to changes.

The saturation index (SI) value less than zero indicates that the phase is soluble, with an SI greater than zero, the phase forms an insoluble compound. Thermodynamic calculation showed that microbial processes contribute to the formation of new insoluble mineral phases of iron, plutonium, and uranium. Formation of ferrous phases in bottom sediments such as pyrite, wurtzite and hydrotroillite will lead to formation of sedimentation-sorption biogenic mineral barrier. The addition of sulfate as a source of sulfur is important. The addition of clay will lead to the formation of an additional sorption layer for caesium.

### 3.6. Verification of Mineral Phase Formation Calculations

#### SEM EDX Sludge Analysis after Biostimulation

When analysing the sludge under study by scanning electron microscopy, the formation of biogenic sediments of various kinds was detected (Table 5).

In samples (B, C, D ) (Figure 10), the presence of iron sulphide of different degrees of crystallization were detected. In samples C, biogenic crystallites of the correct cubic shape for pyrite were found, while in sample B, which is a mixture of bottom sediment and sulfate, the formed iron sulphide was poorly crystallized, indicating that the microbial formation of crystals was not complete. In sample C, with the addition of Ap to the sludge, the formation of pyrite crystals was observed to be the most regular in terms of shape. In sample D, which is a mixture of Ap and sulfate, not only high levels of pyrite content but also significant amounts of new biogenic phases in the form of calcite and calcium phosphate were found. Thus, it can be concluded that Ap, as a source of phosphorus and nitrogen, is beneficial for the biogenic formation of iron-sulphide mineral phases.

## 4. Discussion

Due to the Chernobyl accident uranium fission products (mainly cesium and strontium) and products of its activation (plutonium, americium, neptunium, etc.) were released into the environment. During the analysis of samples from the river Upa, situated in the affected zone, the bottom sediments did not show any significant radioactive contamination. The exception was a small concentration of ^137^Cs (109 Bq/kg). It is likely that the area we examined was not contaminated with actinides in the 1986 accident. It is also possible that the actinides trapped in the sediments mineralized and their concentrations in organic matter became below the sensitivity limit of the equipment.

In this work, we evaluated the involvement of the microbial communities of phytoplankton and bottom sediments in the removal of radionuclides from the aqueous phase to the bottom sediments and their subsequent mineralization. In general, a rather diverse composition of phytoplankton organisms, including representatives of filamentous and unicellular cyanobacteria, unicellular and multicellular green algae, diatomic algae and a number of organotrophic bacteria can be noted. However, its productivity in Upa low-salinity river samples was not high and increased significantly when stimulated by nitrogen and phosphorus sources. In case of active phytoplankton development, the sedimentation of all radionuclides to bottom sediments has been established. In case of non-flowing highly productive water bodies the main condition for hypereutrification is high summer temperatures.

One of the main factors for the reliable immobilization of radionuclides in bottom sediments is the formation of anaerobic conditions due microbial activity and the presence of some components leading to the formation of low soluble phases. The main reason for the Eh decrease in the bottom sediments is the aerobic organic carbon oxidation by the organotrophic bacteria, which were found in the samples (representatives of the families Xantomonadales, Pseudomonadales, Sphingomonadales, Oscilospirales, Clostridiales, Micrococcales etc.). Eh reduction in bottom sediments triggers a cycle of anaerobic microbial processes including anaerobic oxidation of organic matter, sulphate reductions, iron reduction, also. In the case of Upa River sediments, these processes are carried out by sulphate-reducing bacteria of the order Desulfobacterota, genera Desulfotomaculum, Desulfosporomusa, Desulfosporosinus, Thermodesulfobium. The last two processes contribute to the formation of authigenic ferrous and sulphur-ferrous mineral phases in bottom sediments. The formation of biogenic minerals such as hydrotroillite, pyrite, and goethite was predicted by our thermodynamic modeling and detected by scanning microscopy sediment analysis. Earlier it was shown by us and other colleagues that sulphide-ferrous and ferrous mineral phases (sometimes with low crystallinity) are an important factor for immobilisation of uranium and plutonium in the environment [65,66,67,68,69,70]. Another important factor for radionuclide immobilization in bottom sediments is the presence of phosphates, which we applied to stimulate phytoplankton development, in the case of Upa River samples. Uranium and plutonium, as well as other actinides are characterized by the presence of significant amounts of phosphate minerals, including those of biogenic origin [63,71,72,73].

The formation of insoluble uranium phases can be facilitated by microbial processes of uranium respiration including sulfate-reducing bacteria [74,75]. It is important to note that members of the Pseudomonadacaeae and Geobacteriaceae families (genera Pseudomonas and Geobacter) found in the bottom sediments sample are known to be able to use uranium in the dissimilation process [76,77].

In this work the formation of carbonate phases due to microbial oxidation of organic matter was noted. Formation of calcium (calcite), magnesium, iron (siderite) manganese (rhodochrosite) carbonates in bottom sediments can lead to the co-precipitation of actinides, such as uranium, as shown in [78,79,80,81].

Thus, triggering microbial processes by nitrogen and phosphorus sources can accelerate the self-cleaning of a water body from radionuclides and, over several growing seasons, transfer them into bottom sediments. It is possible to predict the formation of an organic-mineral geochemical barrier in bottom sediments in the presence of sulfate and phosphate under anaerobic conditions due to the activity of anaerobic microflora. The formation of anaerobic sulfide and oxide ferrous mineralized barrier in bottom sediments may act for both: radionuclide immobilization increasing [82] and its oxidation preventing [67,82,83]. Stimulation of hypereutrification is most beneficial for non-flooded radioactive waste storage ponds, which are subject to preservation. Purification by an inexpensive aqueous phase method will allow discharge to the surrounding water bodies and further concentration of the bottom sediments for their disposal.

## 5. Conclusions

Experiments were conducted with samples obtained from the Upa River. Water affected after the Chernobyl accident shows that phytoplankton growth is an important mechanism of self-purification of a water body from uranium, strontium, and plutonium. Stimulation of phytoplankton processes by nitrogen and phosphorus sources can result in the accelerated transfer of Pu, U, and Sr into bottom sediments during several vegetation periods. The development of cyanobacteria of the genus Planktothrix allows to expect a high biological productivity in the system [84] including the growth of organotrophic aerobic and anaerobic microflora. Further oxygen content reduction in water during hypereutrophication will hasten the deposition of radionuclides with a variable oxidation state in their reduced low-soluble forms.

An important aspect of reliable radionuclide immobilization in bottom sediments is the course of anaerobic microbial sedimentation processes in the presence of iron and sulfate. Anaerobic microbial community with dominance of sulphur and iron cycle bacteria (Genera Desulfobacterota, Desulfotomaculum, Desulfosporomusa, Desulfosporosinus, Thermodesulfobium, Thiomonas, Thiobacillus, Sulfuritallea) which can be activated upon addition of sulphates was found in the samples of Upa river bottom sediment. Thus, directed hyperactivation of a water body with radionuclide contamination may be a promising in situ technology for reducing the technogenic load of both natural and artificial water bodies. This method can be used to accelerate the clarification of the water phase of engineering objects (storage pools prior to conservation), allowing the treated water to be discharged into the open hydro-network. 

## Figures and Tables

**Figure 1 biology-12-00010-f001:**
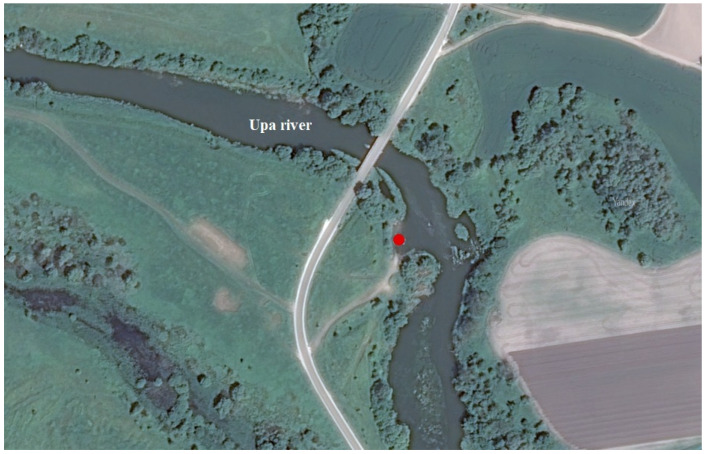
Sampling site (Upa River floodplain, Tula Oblast).

**Figure 2 biology-12-00010-f002:**
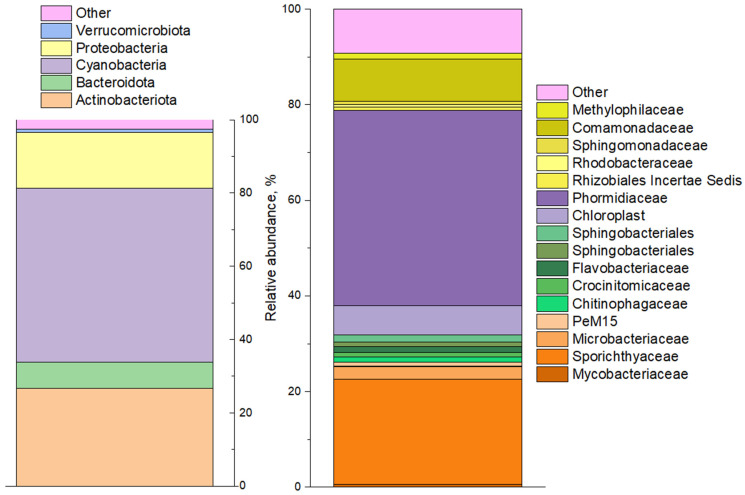
Prokaryotic community diversity profile of the phytoplankton community by 16SrRNA genes (phylum level shown on left, and familia level shown on right).

**Figure 3 biology-12-00010-f003:**
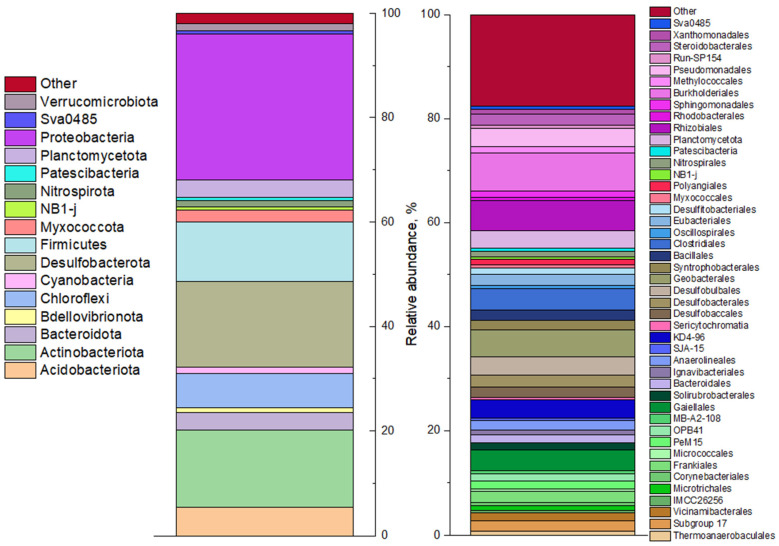
Profiling the microbial community of an anaerobic sludge sample by 16S rRNA genes (phylum level shown on the left and familia level on the right).

**Figure 4 biology-12-00010-f004:**
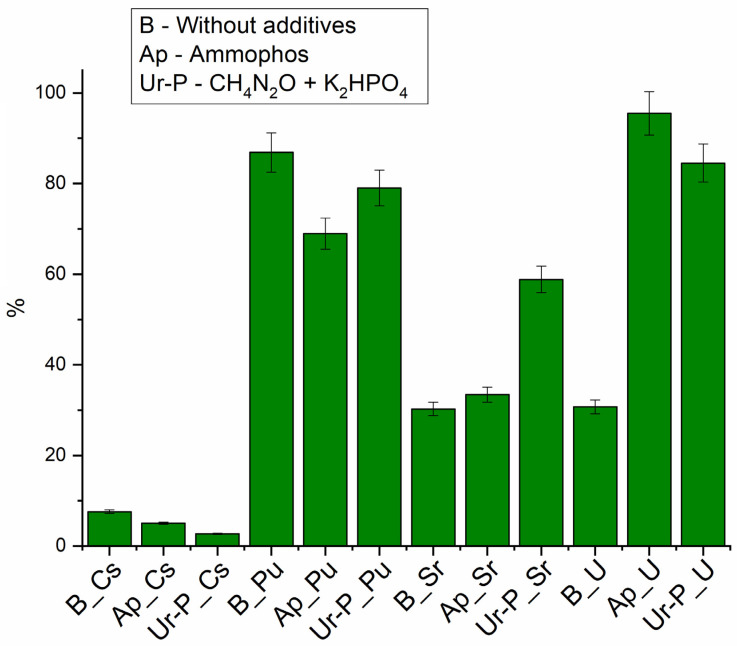
Efficiency of radionuclide removal from the liquid phase, 36 days.

**Figure 5 biology-12-00010-f005:**
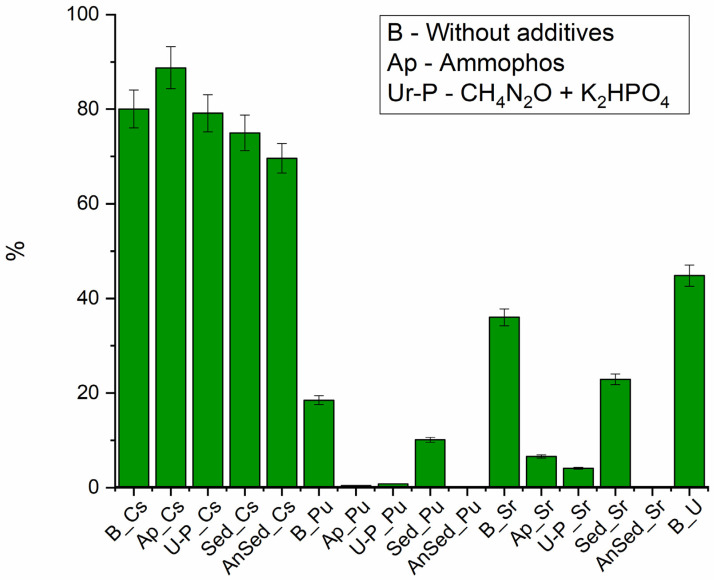
Desorption efficiency of radionuclides from biomass, Sed—sediments in oxic conditions, AnSed—sediments in anoxic conditions.

**Figure 6 biology-12-00010-f006:**
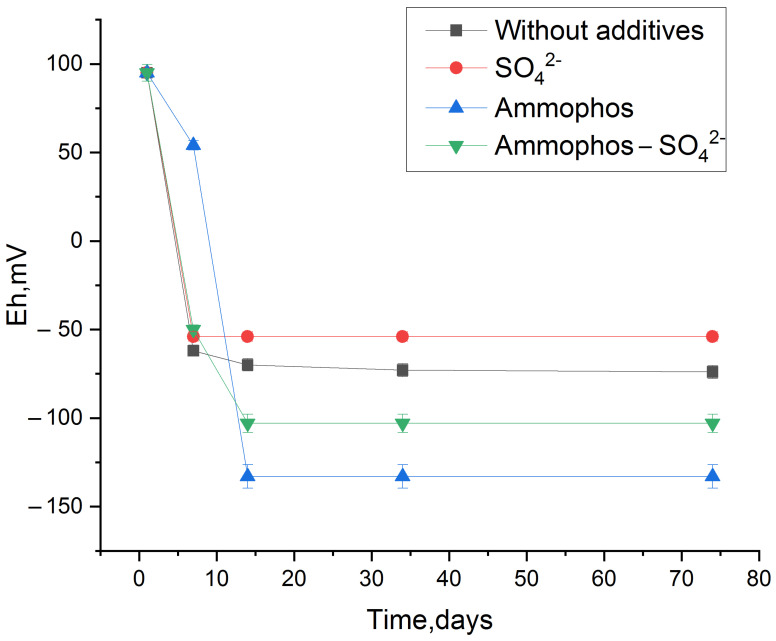
Reduction in redox potential of the bottom sediment system when stimulated with Amophos and sulfate.

**Figure 7 biology-12-00010-f007:**
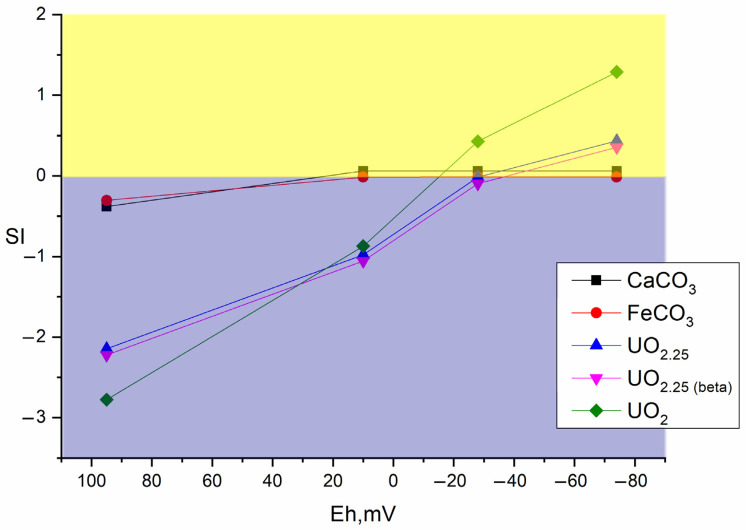
Distribution of main mineral phases before and after microbial processes in the sample without additives.

**Figure 8 biology-12-00010-f008:**
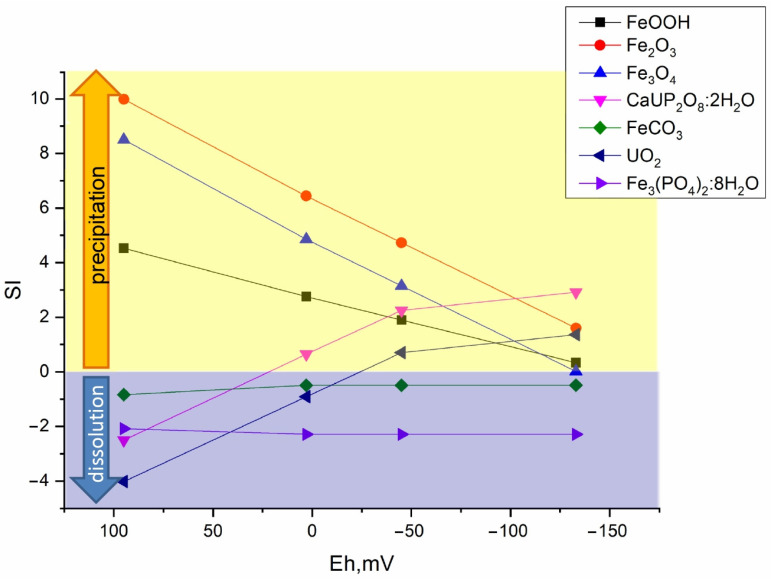
Distribution of main mineral phases before and after microbial processes in the Ammophos sample.

**Figure 9 biology-12-00010-f009:**
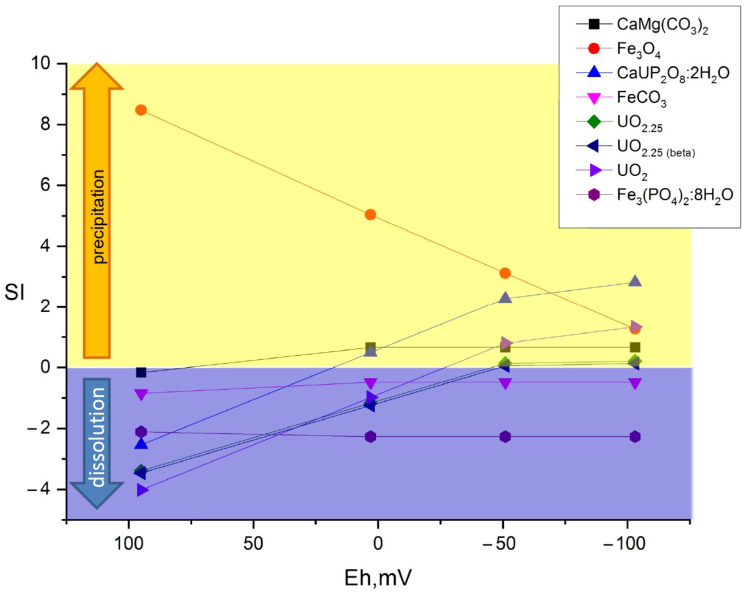
Distribution of main mineral phases before and after microbial processes in the sulfate and Ammophos sample.

**Figure 10 biology-12-00010-f010:**
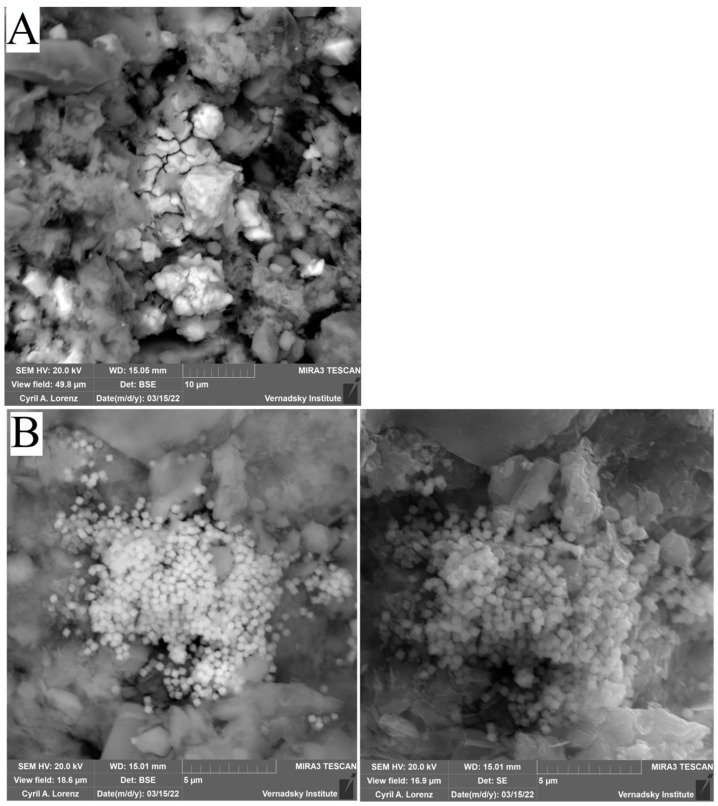
Verification results of thermodynamic calculations obtained by electron scanning microscopy with elemental analysis of the systems: (**A**) SO_4_; (**B**) Ammophos; (**C**) Ammophos + SO_4_; (**D**) Ammophos + SO_4_.

**Table 1 biology-12-00010-t001:** Ion concentrations in the water sample, mg/L.

Ions	Values
HCO_3_^−^	92.1 ± 2.3
Cl^−^	92.7 ± 2.32
Mg^2+^	5.8 ± 0.15
Na^+^	67.9 ± 1.70
Ca^2+^	4.5 ± 0.10
K^+^	11.8 ± 0.30
NO_3_^−^	1.7 ± 0.04
SO_4_^2−^	2.9 ± 0.07

**Table 2 biology-12-00010-t002:** Trace element concentrations in the water sample, µg/L.

Microcomponents	Values
Li	350 ± 8.8
B	230 ± 5.8
Al	1400 ± 35
Si	1200 ± 30
P	380 ± 9.5
Mn	4200 ± 105
Fe	3200 ± 80
Co	10 ± 0.3
Ni	290 ± 7.3
Cu	17 ± 0.4
Zn	49 ± 1.2
Rb	3.2 ± 0.1
Sr	550 ± 13.8
Y	4600 ± 115
Zr	3100 ± 77.5
Cs	520 ± 13
Ba	360 ± 9
Nd	11 ± 0.3
Pb	2.3 ± 0.1
U	0.62 ± 0.02

**Table 3 biology-12-00010-t003:** Chemical composition of the bottom sediments measured via X-ray fluorescence analysis, % mass.

LOI *	Na_2_O	MgO	Al_2_O_3_	SiO_2_	K_2_O	CaO	TiO_2_	MnO	Fe_2_O_3_	P_2_O_5_	S
71.5	1.20	0.96	2.60	17.56	1.20	0.46	0.570	0.06	2.84	0.08	0.03
±	±	±	±	±	±	±	±	±	±		
0.35	0.006	0.004	0.01	0.087	0.06	0.023	0.028	0.003	0.014	0.004	0.0015

* LOI—loss on ignition.

**Table 4 biology-12-00010-t004:** Increase in phytoplankton biomass after adding stimulant additives.

	No Additives		Ammophos		Urea + K_2_HPO_4_	
	Biomass, mg/L	Number of Morphotypes, pcs.	Biomass, mg/L	Number of Morphotypes, pcs.	Biomass, mg/L	Number of Morphotypes, pcs.
K	20 ± 2	9	75 ± 7.5	14	165 ± 16.5	15
U	9 ± 0.9	1	60 ± 6	5	75 ± 7.5	3
Pu	7 ± 0.7	1	50 ± 5	4	125 ± 12.5	6
Cs	8 ± 0.8	4	25 ± 2.5	3	50 ± 5	4
Sr	9 ± 0.9	3	35 ± 3.5	2	75 ± 7.5	2

K—experiment with lighted water sample without additives.

**Table 5 biology-12-00010-t005:** Composition of mineral phases in microphotographs. Instrumental error 0.05 mas%.

Photo	O	Mg	Al	Si	P	S	K	Ca	Fe
A (SO_4_)	14.9	-	0.8	1.9	-	47.7	0.2	0.3	42.7
B (Ap)	32.6	-	1.3	4.1	0.8	31.4	0.14	1.5	35.7
C (Ap + SO_4_)	41.3	0.4	0.6	0.8	-	-	-	31.4	0.3
D (Ap + SO_4_)	22.9	-	0.4	1.2	1.2	41.8	-	1.7	30.6

## Data Availability

Not applicable.

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
