# Peer review of "Biogeochemical Factors of Cs, Sr, U, Pu Immobilization in Bottom Sediments of the Upa River, Located in the Zone of Chernobyl Accident"

_biology, 2022, doi:10.3390/biology12010010_

Round 1
Reviewer 1 Report
1. The genus of microorganisms and all scientific names should be written in italics, check abstract conclusions and references.
2. In materials an methods is written Upa river and Úpa river, correct it.
3. Some references are written in capital letters, change to homogenize the format.
Author Response
Thank you very much for the time devoted to our work. All comments have been corrected.

Reviewer 2 Report
The manuscript deals with an important topic. The reduction of risk following from nuclear disasters (Chernobyl a. p. s, Fukushima a. p. s.) is highly needed. The methods described in the manuscript can reach radionuclides biogeochemical immobilisation via support of phytoplankton activity by low - cost methods. The manuscript is thoroughly prepared, I have no principal or technical remarks.
Author Response
Thank you very much for the time devoted to our work.
Reviewer 3 Report
I have found the manuscript well written and of real interest following the possible development of atomic energetics. The subject is inciting and the results worth publishing. The introduction was written in a very didactic style which should be appreciated.
My general recommendation: accept after the authors will include the corresponding uncertainties as well as error bars.
Some small remarks can be found on the attached annotated file.

Author Response
Thank you very much for your attention to our manuscript.

Reviewer 4 Report
Comments to Authors
Manuscript Number: biology-2062330
Title: Biogeochemical Factors of Cs, Sr, U, Pu Immobilization in Bottom Sediments of the Upa River, Located in the Zone of Chernobyl Accident
Recommendation: Major Revision
Line 62: It should be “as well as intermediate-lived ones”.
Line 63-65, 80-82, 366-369: These sentences were too complex, whose meaning were not clear for readers. Please modify them.
Line 77: “Eh”. Please give the full name before abbreviation.
Line 77-78: “Radionuclides can bind with suspended particles, can be precipitated and fixed in bottom sediments.” Lack of link words between two short sentences. Please check it.
Line 80-85: Why were there only two small paragraphs here? The relevant research background was insufficient. Please check it.
Line 91: This sentence was strange. The grammar maybe incorrect.
Line 114: “53.954297° X, 37.158518° X”. The “N, S, E, W” should be used to represent the direction of longitude and latitude.
Line 162-163: The units of “Csph” and “Clph” should be given.
Line 201, 203, 204: The unit of temperature was strange. It should be “K”.
Line 178-222: There are too many short paragraphs, please try to combine them.
Line 216-222: Much more details about the DNA extraction, PCR amplification and gene sequencing should be given. Moreover, the raw sequencing data should be offered or deposited to professional institutes, which can be accessed online.
Line 238: “б”?
Line 239: It should be “+95 mV”?
Line 241: Please use three-line table and optimize table form. The current form is casual and ugly.
Line 243-244: What were the maximum permissible concentrations? Please be specific.
Line 279: Repeated “representatives” should be deleted.
Line 284, 300: Within Figs. 2 and 3, the classification level should be given. At the phylum level? Or at the genus level? Each figure has two sub-figures (left and right). The authors need to explain the meaning of the sub-figures.
Page S2: Within Fig. S2 caption, it should be “16S rRNA”.
Line 241, 251, 316, 418: Regarding all tables, the three-line form is required. And, the form of data should be modified. The decimal point should be “.” not “,”.
Line 338-340: This part is too short. Please combine it with section 3.4.2, or add more discussion in it.
Line 364: Much more explanation should be given in this part.
Line 429: Regarding the Section 3. Results and discussion, much more discussions should be provided to take all results together and to reach some clear conclusions. Moreover, the environmental implications should be discussed in this section.
Line 446-449: This part should be transferred to the discussion part. The conclusion should summary the whole research briefly and put forward the future study direction.
Line 454: “6. Patents”? Confusing.
Author Response

(The authors gave the same response as above.)

Round 2
Reviewer 4 Report
I have read the authors' responses to my comments and am satisfied by the modifications made. Some minor suggestions: The decimal point should be “.” not “,” in Table 3. Moreover, some expressions should be checked agnain before accepted.Author Response
Dear reviewer, thank you for the time devoted to our article. All your comments have been taken into account and marked in the text.
